# Biocompatibility Assessment of Polylactic Acid (PLA) and Nanobioglass (n-BG) Nanocomposites for Biomedical Applications

**DOI:** 10.3390/molecules27113640

**Published:** 2022-06-06

**Authors:** Jorge Iván Castro, Carlos Humberto Valencia Llano, Diego López Tenorio, Marcela Saavedra, Paula Zapata, Diana Paola Navia-Porras, Johannes Delgado-Ospina, Manuel N. Chaur, José Hermínsul Mina Hernández, Carlos David Grande-Tovar

**Affiliations:** 1Laboratorio SIMERQO, Departamento de Química, Universidad del Valle, Calle 13 # 100-00, Cali 76001, Colombia; jorge.castro@correounivalle.edu.co (J.I.C.); manuel.chaur@correounivalle.edu.co (M.N.C.); 2Grupo Biomateriales Dentales, Escuela de Odontología, Universidad del Valle, Calle 4B # 36-00, Cali 76001, Colombia; carlos.humberto.valencia@correounivalle.edu.co (C.H.V.L.); dlopeztenorio@hotmail.com (D.L.T.); 3Grupo de Polímeros, Facultad de Química y Biología, Universidad de Santiago de Chile, USACH, Santiago 9170020, Chile; alecram.saavedra@gmail.com (M.S.); paula.zapata@usach.cl (P.Z.); 4Grupo de Investigación Biotecnología, Facultad de Ingeniería, Universidad de San Buenaventura Cali, Carrera 122 # 6-65, Cali 76001, Colombia; dpnavia@usbcali.edu.co (D.P.N.-P.); jdelgado1@usbcali.edu.co (J.D.-O.); 5Grupo de Materiales Compuestos, Escuela de Ingeniería de Materiales, Facultad de Ingeniería, Universidad del Valle, Calle 13 # 100-00, Cali 760032, Colombia; jose.mina@correounivalle.edu.co; 6Grupo de Investigación de Fotoquímica y Fotobiología, Universidad del Atlántico, Carrera 30 Número 8-49, Puerto Colombia 081008, Colombia

**Keywords:** antimicrobial, biocompatibility, cell viability, histology, nanobioglass, nanocomposites, polylactic acid

## Abstract

Scaffolds based on biopolymers and nanomaterials with appropriate mechanical properties and high biocompatibility are desirable in tissue engineering. Therefore, polylactic acid (PLA) nanocomposites were prepared with ceramic nanobioglass (PLA/n-BGs) at 5 and 10 wt.%. Bioglass nanoparticles (n-BGs) were prepared using a sol–gel methodology with a size of ca. 24.87 ± 6.26 nm. In addition, they showed the ability to inhibit bacteria such as *Escherichia coli* (ATCC 11775), *Vibrio parahaemolyticus* (ATCC 17802), *Staphylococcus aureus* subsp. aureus (ATCC 55804), and *Bacillus cereus* (ATCC 13061) at concentrations of 20 *w*/*v*%. The analysis of the nanocomposite microstructures exhibited a heterogeneous sponge-like morphology. The mechanical properties showed that the addition of 5 wt.% n-BG increased the elastic modulus of PLA by ca. 91.3% (from 1.49 ± 0.44 to 2.85 ± 0.99 MPa) and influenced the resorption capacity, as shown by histological analyses in biomodels. The incorporation of n-BGs decreased the PLA crystallinity (from 7.1% to 4.98%) and increased the glass transition temperature (T_g_) from 53 °C to 63 °C. In addition, the n-BGs increased the thermal stability due to the nanoparticle’s intercalation between the polymeric chains and the reduction in their movement. The histological implantation of the nanocomposites and the cell viability with HeLa cells higher than 80% demonstrated their biocompatibility character with a greater resorption capacity than PLA. These results show the potential of PLA/n-BGs nanocomposites for biomedical applications, especially for long healing processes such as bone tissue repair and avoiding microbial contamination.

## 1. Introduction

Nanocomposite scaffolds based on polymeric matrices that simulate the complex morphology of tissues with biocompatible materials have become an area of interest in medicine [1]. The scaffolds are a temporary skeleton for progenitor cell attachment, proliferation, and differentiation in a 3D architecture that resembles harvested tissue. This tissue provides pathways for the entry of nutrients and the exit of metabolic products [2]. The goal is, in the end, that the biological environment reabsorbs the scaffold while the tissue is repaired. However, despite having a 3D structure, many biomaterials do not have all the characteristics to stimulate cell regeneration. In this sense, many researchers have developed hybrid materials combining the benefits of each material in the mixture to regenerate tissue and even cure diseases such as cancer [3].

Generally, nanostructured materials produce three-dimensional surfaces of scaffolds with nanometric characteristics [4]. Within these materials are nanoparticles [5], nanofibers [6], nanocomposites [7], and nanosheets [8], which increase cell interaction and promote integration into the host tissue [9]. However, biocompatibility is limited by the architecture of the scaffold, as it provides an optimized microenvironment for synthesizing new tissue. This architecture allows for the flow of micronutrients and metabolites between cells and the surrounding extracellular matrix (ECM). Sometimes, it is necessary to add specific fillers of bioactive materials to the polymer mixture to imitate the chemical, mechanical, and thermal properties and their biocompatibility [10].

A bioactive material is defined as a material that can form a layer of hydroxyapatite (HA) to bind tissue or bone [11,12]. Nanobioglasses (n-BGs) are bioactive inorganic materials useful as fillers in preparing biopolymers scaffolds [13]. On the other hand, it has been shown that the presence of n-BGs does not provoke a foreign body response on the surface when it comes into contact with biological fluids, which has raised scientific interest for applications in tissue engineering. Consequently, n-BGs have the advantage of promoting enhanced angiogenesis and upregulation of specific genes that control the cell cycle of osteoblasts [14].

Different types of n-BGs have been synthesized to tune scaffolds’ in vitro chemical properties, such as 45S5 and 1393BG [10]. The 1393BG is approved by the European Union for in vivo experiments and comprises a weight composition of SiO_2_ 53%, Na_2_O 6%, CaO 20%, P_2_O_5_ 4%, K_2_O 12%, and MgO 5%. These n-BGs contain an amount of silicon above 50% with network modifiers such as K_2_O and MgO [15,16]. However, 45S5-type n-BGs with a composition of 46.1% SiO_2_, 24.4% Na_2_O, 26.9% CaO, and 2.6% P_2_O_5_ are preferred due to their many biomedical applications [17,18]. However, the main disadvantage is the proper porosity control. It requires special conditions for sintering, which can be affected by the phenomenon of nucleation and the growth of crystalline phases occurring during heating [19,20,21].

Of particular interest in recent years has been the combination of the properties of n-BGs with biopolymers in scaffolds to obtain materials with mechanical strength, porosity, resistance to thermal and hydrolytic degradation, and stimulation for osteogenesis. The remarkable success is the increased chemical properties provided by the polymer and the bioactivity introduced by the n-BGs. Within the polymers, α-polyesters with n-BGs have been used due to their excellent mechanical properties, low toxicity, and predictable biodegradation kinetics. Among the α-polyesters, polylactic acid (PLA) [22] has been considered a critical polymeric matrix due to its excellent biocompatibility, high crystallinity, hydrophobicity, processability, and degradation in physiological environments [23,24]. However, the neat PLA has disadvantages related to its cellular response and mechanical properties. On the other hand, PLA/n-BGs composites have been synthesized at different concentrations of n-BGs by thermally induced phase separation (TIPS), considering n-BGs of the 45S5BG type, which exhibited an anisotropic structure similar to a ladder due to the solid–liquid phase separation that occurs during scaffold formation.

Fibrous scaffolds have been synthesized for bone regeneration through the electrospinning technique between PLA and n-BGs, with sizes between 320–550 nm. The nanofiber, after heat pressing, produces dense nanocomposites that induce the formation of a hydroxycarbonate apatite layer on the surface under a simulated physiological environment. Despite the reported applications of PLA/n-BGs nanocomposites, no studies evaluating the biocompatibility under in vivo conditions for nanocomposites of PLA/n-BGs have been reported. For this reason, the biocompatibility of n-BGs with HeLa cells and with biomodels was assessed. The evaluation results demonstrated the potential in tissue engineering of PLA/n-BGs by combining the properties of n-BGs and PLA, improving the biocompatibility and the thermal/mechanical properties of PLA/n-BGs scaffolds.

## 2. Results and Discussion

### 2.1. Characterization of the n-BGs

#### 2.1.1. Fourier Transform Infrared Spectroscopy (FT-IR)

Figure 1 shows the FT-IR spectrum of n-BGs. Asymmetric strain bands and bending of the Si-O-Si bonds at 1069 and 586 cm^−1^ were observed. The bending band centered at 802 cm^−1^ corresponds to the Si-O bond. Finally, the band at 586 cm^−1^ corresponds to the antisymmetric bending of the P-O bond [5]. However, it should be noted that the P-O bond band overlaps with the characteristic Si-O bands. Additionally, the symmetric strain band corresponding to the carbonate ion and the band corresponding to the free OH groups of water are found at 1424 and 1591 cm^−1^. These observations are similar to previous works [5,25].

#### 2.1.2. X-ray Diffraction (XRD) of n-BGs

X-ray diffraction (XRD) allows for quantitative and qualitative measurements of the crystalline phases of compounds from their diffraction planes. Figure 2 shows the XRD spectrum for the n-BGs, which shows that the amorphous phase predominated over the crystalline phase [5]. However, between the phases of the n-BGs and their crystalline phase, a broad peak emerged around the 2θ angle of 29°, indicating a higher population density and a smaller distance from any central atom [26]. The higher density of the n-BGs can be attributed to network modifications within the empty polyhedra of the loose silicate network. This same crystalline phase has been observed in other works [5,26].

#### 2.1.3. Thermal Analysis for the n-BGs

The results of the TGA of the n-BGs are presented in Figure 3. Thermal degradation is divided into three main stages. The first is around 100 °C, attributable to the loss of water present in the material. The second stage begins at about 400 °C, attributable to the loss of -OH groups. Finally, a slight weight loss is observed at 600 °C, attributed to the beginning of crystallization [27].

#### 2.1.4. Transmission Electron Microscopy (TEM)

Figure 4A presents the Transmission Electron Microscopy (TEM) analysis of the n-BGs obtained by the sol–gel method with an average particle size of ca. 24.87 ± 6.26 nm. The shape of the nanoparticles was primarily spherical. However, it can also be seen that there were agglomerations of nanoparticles, something expected in this type of methodology preparation of n-BGs [28]. To determine the particle size, the diameter of 100 nanoparticles was taken using the Image J program, with which the size distribution histogram of Figure 4B was constructed.

### 2.2. Characterization of PLA/n-BGs Nanocomposites

#### 2.2.1. Fourier-Transform Infrared Spectroscopy (FT-IR)

Figure 5 shows the IR spectrum corresponding to the n-BGs/PLA scaffolds. All samples exhibited molecular vibrations for the different functional groups in the IR for PLA and the n-BGs.

For the PLA, we can find the symmetric and asymmetric tension band of the alkyl group -CH between 2800–3200 cm^−1^ and the bending band of the alkyl groups -CH, -CH_3,_ and the stretching of the C-O-C group between 1000–1500 cm^−1^. Finally, the stretching corresponding to the carbon chain of the PLA and the oscillating band of the CH_3_ group was between 830–960 cm^−1^ [10]. The PLA/n-BGs scaffolds at 5 wt.% and 10 wt.% had the PLA bands. However, the characteristic bands of the n-BGs at 1182 and 757 cm^−1^ corresponding to the symmetric and asymmetric stretching of the Si-O-Si groups were evident.

Additionally, intensity changes in the bands between 1000–1500 cm^−1^ corresponding to the CH, CH_3_, and COC groups from PLA were observed. As the concentration of the n-BGs increased, the band at 1456 cm^−1^ of the asymmetric deformation of the CH_3_ group decreased, probably affected by changes in the crystallization behavior of the PLA [29]. Finally, the bands between 1100 and 1300 cm^−1^ of the C-O-C vibrational modes and the asymmetric oscillating vibrations -CH_3_- overlapped with the Si-O-Si band of the n-BGs [5,29].

#### 2.2.2. Thermal Analysis of PLA/n-BGs Nanocomposites

Thermogravimetric analysis (TGA) aims to determine the stability of the material as the temperature increases. Figure 6A shows the thermograms of the neat PLA and the nanocomposites when heated to 900 °C under a nitrogen atmosphere. The 10% decomposition temperature (T_10_) and maximum decomposition (T_max_) are shown in Table 1. The T_10_ increased ca. 44.2% with the incorporation of 5 wt.% n-BGs, although it did not increase with further incorporation of n-BGs. This increase implies that the thermal stability of the PLA increased with the incorporation of n-BGs due to the hydrogen bonds between the carbonyl group of the PLA and the hydroxyl groups of the n-BGs and van der Waals interactions [30]. The increase in T_10_ is a good indication of better thermal stability. It has been reported that PLA depolymerization occurs at 250–320 °C, while for PLA/n-BGs nanocomposites, a lower weight loss is observed between 100 and 300 °C [31]. 

Thermal stability occurs because the silicate layers of n-BGs act as a barrier to incoming gases and gaseous by-products during thermal degradation, increasing the strength of the material and acting as a thermal insulator [4]. It has been previously reported that the degradation temperature depends mainly on the n-BGs, which is related to the hydrolysis conditions of the polymer since it interferes with the heat transfer of the components [27,32]. On the one hand, the T_max_ did not change with incorporating n-BGs into the PLA. The incorporation of n-BGs stabilized the PLA only at low temperatures. Still, it did not affect the maximum degradation temperature since a change in the chemical structure of the PLA was not generated.

On the other hand, differential scanning calorimetry (DSC) analyses allow for the study of polymer thermal transitions, determining glass transition (T_g_), melting (T_m_), crystallization (T_cc_) temperatures, and changes in enthalpy of these processes when the temperature is increased [33]. Figure 7 and Table 1 show the values of the T_g_, T_cc_, T_m1_, and T_m2_ of the neat PLA and the nanocomposites PLA/n-BGs 5 wt.% and PLA/n-BGs 10 wt.% obtained in the second heating. As seen in Table 1, the T_g_ of the PLA increased from 53 °C to 63 °C with the incorporation of n-BGs at 5 wt.% due to the presence of nanoparticles between the polymer chains, which decreased the polymeric chains’ movement [4]. PLA presents this endothermic peak thanks to the chain relaxation in the structure and the freedom of movement they offer.

The excellent dispersion of the n-BGs in the polymeric matrix favored the PLA and n-BGs London and hydrophobic interactions. For the PLA, it is possible to observe the T_cc_ as an exothermic peak due to the crystallization of the polymer chains during cooling and two endothermic transitions due to the melting of the PLA crystalline domains (T_m1_ and T_m2_). An important consideration is the similar enthalpies of the two contiguous processes, indicating a low crystalline character of the PLA used [34]. The presence of two T_m_ indicates polymorphism of the different crystal structures present in the PLA mixture. 

On the other hand, T_cc_ and T_m1_ were not observed for nanocomposites, suggesting a decrease in crystallinity. The incorporation of the nanoparticles might have affected the nucleation of the PLA chains, which ultimately affected crystallization. Table 1 shows a reduction in the percentage of crystallinity Χc, from 7.8% to 5.0%, with the incorporation of 5 wt.% of n-BGs. However, the T_m2_ practically did not change, indicating that the fusion of the significant domain of the PLA was still present.

#### 2.2.3. XRD of PLA/n-BGs Nanocomposites

The XRD method is based on the ability of the material to diffract X-rays. The diffraction pattern is dependent on the crystalline phases [35]. Figure 8 shows the XRD of the PLA/n-BGs 5 wt.% and PLA/n-BGs 10 wt.% nanocomposites. 

Crystallographic data for the PLA demonstrate its orthorhombic nature with well-defined peaks at specific 2θ values (15° (011), 16° (101), 19° (012), 22° (013), 29° (004), and 31° (021)) [36,37]. For the nanocomposites, we observed broad peaks at 2θ = 27°, 29°, and 31° as a result of crystal structure changes upon n-BGs incorporation. The 2θ = 15°, 17°, 19°, and 22° peaks were related to the crystal structure of the PLA.

Previous works have shown that the inclusion of n-BGs promotes the formation of hydroxyapatite on the surface of the material through the stimulation of nucleation processes [14]. Therefore, the decrease in crystallinity for the PLA/n-BGs nanocomposites 5 wt.% and PLA/n-BGs 10 wt.% of 71.8 and 71.1%, respectively, was because nanocomposites do not have compact crystal lattices, with nucleation predominating on the material’s surface [14,38].

#### 2.2.4. Morphological Analysis of neat PLA and PLA/n-BGs Nanocomposites

Scanning electron micrographs allow the surface morphology of the material to be elucidated through an electron microscope. Figure 9 shows the micrographs of the neat PLA, PLA/n-BGs 5 wt.% and PLA/n-BGs 10 wt.%. For the PLA, a rough heterogenous morphology was observed. The nanocomposites’ structure was characterized by a grid-like texture with thick walls that separate large permeable cavities from the sample.

Additionally, it was observed that the increase in the concentration of the n-BGs did not affect its internal sponge skeleton-like structure, which is characteristic of materials containing n-BGs. However, a higher filling of the interstices was seen in the nanocomposite with a higher proportion of n-BGs [39]. In fact, by having an organic compound such as PLA, the scaffolds had a more extensive pore network due to decomposition gases from the evaporation of the organic matrix. The presence of the PLA then promoted a rough surface as it did not present a defined microstructure, which supported the diffusion of fluids allowing the adhesion of proteins and cells [14,39].

According to Belluci et al. [14], scaffolds such as those synthesized in this work are defined as “shell scaffolds”. These scaffolds are characterized by a highly interconnected and open internal porosity and a compact yet permeable external surface. The outer surface, which resembles a shell, acts as an exoskeleton for the entire scaffold and facilitates the handling of the scaffold without damage.

#### 2.2.5. TEM Images of the PLA/n-BGs Nanocomposites

The TEM analysis of the PLA/n-BGs nanocomposites is shown in Figure 10. As can be seen, there was a uniform distribution of the n-BGs distributed within the polymeric matrix and some free n-BGs. The agglomerates increased with the n-BGs, confirming their incorporation and homogeneous distribution within the PLA matrix. This high level of dispersion of the nanoparticles demonstrates a good interaction between the PLA and n-BGs, obtained with the nanocomposite synthesis methodology that allowed this homogeneous distribution within the PLA.

#### 2.2.6. Mechanical Tests of PLA/n-BGs Nanocomposites

The biomedical application of biomaterials requires specific mechanical and thermal characteristics due to the surrounding environment in which they will remain until their total degradation. Table 2 shows the mechanical behavior of the neat PLA and PLA/n-BGs nanocomposites. The incorporation of n-BGs at 5 wt.% in the PLA increased ca. 91.3% Young’s modulus, demonstrating a mechanical strengthening effect with incorporating n-BGs. This behavior can be explained by the increase in the material’s stiffness due to the nanoparticles’ large available surface area that facilitates the interactions with PLA chains. This increase in stiffness was also evidenced by the loss of the maximum deformation capacity (from 123.79 ± 50.30% in PLA to 38.46 ± 0.56% for PLA/n-BGs 5 wt.%), causing the efforts to increase significantly [14]. 

The PLA/n-BG nanocomposites presented better mechanical properties than previous works [40] due to the higher molecular weight of the PLA used in this work (200,000 Da) and the reinforcing effect of the n-BGs. In addition, the excellent dispersion of the n-BGs evidenced by TEM allowed for multiple interactions between the nanoparticles with a large surface area and the polymer chains. This significant interaction between the nanomaterial and the polymeric matrix facilitated the transmission of mechanical stress. Similar results have been previously reported for PLA reinforced with other nanomaterials [34].

### 2.3. Biological Tests of n-BGs and PLA/n-BGs Nanocomposites

#### 2.3.1. Antimicrobial Activity of n-BGs against Pathogenic Bacteria

n-BGs generally have low antimicrobial activity [41]. Although the mechanisms by which inhibition occurs have not been precisely defined, it is considered that the activity depends mainly on their physical characteristics (size and surface). The adhesion of microorganisms to n-BGs may influence their growth due to possible membrane damage [41,42]. 

The results showed that the n-BGs induced the antimicrobial activity of gram-positive and gram-negative bacteria at high concentrations (Table 3) using the TTC method, where the red color produced by the dye indicated metabolic activity and microbial growth. *Bacillus cereus* showed the most heightened susceptibility of the strains evaluated according to Adams et al. [41]. In the samples where the test with the TTC method did not show enzymatic activity, the concentration of the residual microorganisms was determined to assess the lethal effect on bacteria, which indicated that only *B. cereus* was wholly inhibited at a concentration of 20 *w*/*v*%. The other strains presented a significantly lower reduction to 1 Log, indicating a sub-lethal effect.

PLA/n-BGs were not tested for their antimicrobial activity in this work. The maximum concentration of n-BGs in PLA/n-BGs that gives it optimal mechanical properties and low cytotoxicity is below the inhibitory concentration found.

#### 2.3.2. MTT Cell Viability Assay with HeLa Cells

The cytotoxicity evaluation of the neat PLA and the nanocomposites was carried out with the salt method (3-(4,5-dimethylthiazol-2-yl)-2,5-diphenyltetrazolium bromide), MTT, one of the most used to carry out a preliminary evaluation of the cytotoxicity of materials. This method can quantify the number of living cells by evaluating cytostatic activity. It is based on living cells converting the MTT component (3-(4,5-dimethylthiazol-2-yl)-2,5-diphenyltetrazolium bromide) into formazan, an insoluble product. Then, the change in color serves as an indication of assessing viable cells. Absorbance measurement at OD590 nm is proportional to the number of live HeLa cells [43]. Figure 11 shows the results of the average MTT viability in the percentage of each formulation and the control.

When comparing the means of the three formulations by Fisher’s test and performing the analysis of variance, it was found that the treatments did not differ significantly with a confidence index (C.I.) of 95%. When comparing the neat PLA and nanocomposite results with those of the positive control (cisplatin), it was found that they did differ significantly with a confidence index of 95% since the MTT viability of cisplatin was 33.6%. The multiple treatment comparisons with the control by Dunnett’s test confirmed the significant differences with a C.I. of 95% of the treatments with the positive control. Interestingly, it was observed that the MTT viability values with HeLa cells remained above 80% and were not affected by the incorporation of n-BGs, indicating the cytocompatibility of these nanocomposites with cells. This result confirms previous reports indicating that n-BGs are not cytotoxic for mammalian cells [44,45] and even improve biocompatibility in vitro [46]. 

PLA has wide applications in tissue engineering due to its recognized biocompatibility [47,48]. n-BGs are also materials widely used in bone tissue engineering due to their biocompatibility and the ability to attract osteogenic cells as they are resorbed when implanted in bone tissue [49]. In addition, their ability to stimulate bone healing has been demonstrated due to their ability to join the bone by depositing a layer of hydroxyapatite on their surface, stimulating cell differentiation [50] and proangiogenic properties [51].

#### 2.3.3. Biocompatibility Test in Biomodels (In Vivo)

Before recovering the samples of the tissues implanted in the biomodels, a macroscopic observation of the intervened area was carried out, following the UNE-EN ISO 10993-6:2017 Standard “Biological evaluation of medical devices, Part 6: Tests related to local effects after implantation”.

Figure 12 corresponds to the dorsal image of one of the biomodels after 30 days of implantation. In all biomodels, a similar appearance was observed in the three implantation periods.

After 30 days of implantation, hair recovery was observed in the three biomodels (Figure 12A). When performing the trichotomy (Figure 12B), it was observed that the surgical lesions were healed with the absence of necrotic tissue or inflammatory or purulent exudate. The healing process was carried out by encapsulating the implanted materials. When dissecting the tissue, the internal surface (Figure 12C) revealed the implanted materials immersed in a more reddish-colored tissue, without a severe inflammatory response or with the presence of pus. The macroscopic image indicated the biocompatibility of the material with the surrounding tissue. 

The macroscopic observation of the intervened biomodels (before the recovery of the samples) showed new hair growth, an external surface of healthy skin without continuity solutions, no exudates, and an internal surface without evidence of purulent or inflammatory exudates. The skin is one barrier that protects the rest of the body. It can regenerate itself in the face of trauma. It goes through the stages of inflammatory reaction, proliferation, and remodeling that allow it to return to the initial state before the trauma. However, excessive trauma or an inflammatory response for prolonged periods can alter the healing process, generate lacerations and ulcers, and produce pus [52].

In addition, it was possible to identify the insertion zones by changes in color and by small bulges with induration, which indicates a fibrous encapsulation of the material. The encapsulation of the material suggests a normal healing process in the presence of a foreign body in which the implanted material is surrounded by fibrous tissue. This tissue will persist until removed, as reported in previous investigations of soft tissue embedded materials [53].

##### Histological Evaluation of Neat PLA Scaffolds

The results of the subdermal implantation of the PLA showed that the material was very stable, remaining in the three intervened biomodels without appreciable changes during the three observation periods (Figure 13).

By staining with Masson’s Trichrome technique, it was observed that at 30 days (Figure 13A), the material was located and surrounded by a network of collagen I fibers (Figure 13D), with an abundant presence of blood vessels and inflammatory cells (Figure 13D).

Collagen fibers persisted at 60 days (Figure 13B) and 90 days (Figure 13C). A fibrous capsule could be seen surrounding the entire implantation site in the image. Each particle was individually wrapped with collagen I fibers, as shown by the typical blue staining of Masson’s trichrome (yellow arrows, Figure 13C,D).

##### Histological Evaluation for the Nanocomposite PLA/n-BGs 5 wt.%

After 30 days of implantation, a significant presence of material samples was observed (Figure 14A), while at 60 days, it was observed that the material was fragmented. The particles started a reabsorption process with the presence of surrounding inflammatory cells (red arrows, Figure 14B). At 90 days (Figure 14C), some small-sized particles with an appearance observed in the 60 days (black oval) were observed. However, there were also groups of particles with a different appearance, which seemed to be in the degradation process (yellow ovals).

As with the PLA sample, the implantation zone was characterized by a capsule of collagen I fibers around the remaining PLA/n-BGs 5 samples (Figure 14D). On the other hand, in the capsule at 60 days, numerous blood vessels (Figure 14E) were evident, which is also an indicator of the healing process.

In Figure 14, it was possible to observe the material’s changes throughout the 90 days of implantation. Initially, there were large size particles (Figure 14A). After 60 days, the material was fragmented by the phagocytic action of the inflammatory cells (Figure 14B). At 90 days, there were particles in the presence of reabsorption and other groups of particles with superficial changes that indicated a degradation process (Figure 14C).

Healing occurred amid a foreign body reaction in which PLA/n-BGs 5 wt.% were encapsulated to allow recovery while reabsorption by inflammatory cells occurred. The implanted material was observed to behave as a biocompatible material within the subdermal tissue.

##### Histological Evaluation for the Nanocomposites PLA/n-BGs 10 wt.%

The healing process of the sample PLA/n-BGs 10 wt.% was very similar to that described for the PLA/n-BGs 5 wt.%. Initially, there were large-sized particles, and the material was surrounded by a fibrous capsule (Figure 15A). At 60 days, the amount of material had decreased, and they presented morphological changes in which the edges tended to become rounded with areas of resorption (Figure 15B). At a higher magnification (100×), it was observed that the particles were surrounded by bundles of collagen fibers with the presence of blood vessels (Figure 15C). Although some material particles persisted at 90 days of implantation, the resorption process was more evident since fewer particles were observed.

On the other hand, as a positive control for a resorbable material, a porcine collagen membrane was implanted at 30, 60, and 90 days. Figure 16 corresponds to the histological image of the tissue implanted with this material. After 30 days, a prominent inflammatory infiltrate was seen at the implantation site. After 60 days, the inflammatory infiltrate was not very appreciable, and at 90 days, it was no longer observed, indicating the complete reabsorption process (Figure 16). 

Figure 16 allows us to observe the healing process of an utterly compatible material (collagen membrane), as it is a material constituted of collagen I and III fibers. It is important to remember that collagen fibers are the structural component of the skin [54]. On the other hand, the inflammatory cells are the ones that carry out the reabsorption–degradation process that culminates in the complete elimination of the implanted material without generating a reaction to a foreign body.

The histological findings related to the resorption of PLA/n-BGs nanocomposites and porcine collagen agree with previous reports. Initially, there is an inflammatory infiltrate between 10 and 30 days, followed by material fragmentation between 30 and 60 days [55]. When the histological findings for the three materials are compared, it is observed that the PLA was the least reabsorbed (degraded), persisting without significant changes at 90 days and always with the presence of the surrounding fibrous capsule. In the case of PLA/n-BGs nanocomposites, it was observed that increasing the percentage of n-BGs accelerated the resorption process. For example, PLA/n-BGs 5 wt.% after 60 days of implantation was observed to be more fragmented and in the process of reabsorption. Additionally, after 90 days, changes were observed in its texture that suggest a degradation process that led to its reabsorption (Figure 14).

The PLA/n-BGs at 10 wt.% seemed to have a faster resorption–degradation process with critical morphological changes. Its fragmentation was observed at 30 days, generating structures with rounded edges at 60 days and decreasing the number of particles at 90 days (Figure 15).

When a biomaterial is implanted, the body’s response will always respond to a foreign body, generating the encapsulation of the inserted material [56,57,58]. This reaction disappears when the material is completely resorbed. In very stable materials, whose resorption takes longer than the evaluated period, the presence of the capsule and an inflammatory infiltrate is considered a good indication of a normal healing process and tissue recovery.

The healing of a tissue (immune response of the organism) goes through several stages. After surgical trauma and implantation, a provisional matrix is formed at the expense of protein absorption on the material; then comes the acute phase. There is an initial inflammatory response lasting approximately one week. If the situation is not resolved in this period, there will be an evolution towards a chronic phase [59].

The chronic phase is established two to five weeks after the initial trauma. It is characterized by an inflammatory infiltrate with the presence of neutrophils, monocytes, lymphocytes, polymorphonuclear cells, and mast cells. If it is not possible to resolve the situation generating the inflammatory response, the formation of foreign body giant cells occurs, a product of the fusion of macrophages, which will be present during the entire period in which the implanted material remains. Subsequently, the fibrous capsule formation occurs, surrounding the implanted material and allowing the cells to degrade it enzymatically with fragmentation and phagocytizing it [59]. 

Granulation tissue is initially formed by a disorganized network of collagen fibers with inflammatory cells and blood vessels. Still, it later matures into more organized, fibrillar, less cellular tissue with the presence of a peripheral fibrous capsule with the replacement of collagen III by collagen I [60]. This process is very similar to our observations in the healing process (30, 60, and 90 days), in which the capsule surrounding the nanocomposites was formed of collagen I fibers with blood vessels.

Finally, there is a resolution phase, in which, before the disappearance of the implanted material, the inflammatory cells and blood vessels decrease while the capsule is reabsorbed. In this research, 90 days after implantation, only the PLA/n-BGs 10 wt.% showed a significant reduction in the implanted material with the persistence of small particles, which prevented the completion of the resolution phase. Hence, networks of collagen fibers were observed surrounding the particles, with the presence of blood vessels.

Interestingly, a higher percentage of n-BGs drove more significant degradation. Probably, a higher porosity due to a higher rate of n-BGs stimulated more increased reabsorption. A higher heterogeneous structure with porosity allows the exchange of fluids with the surrounding cells and the stimulation of adsorption on their surface. In the future, it is crucial to analyze the response of the tissues at the molecular level to check the regeneration of surrounding cells at the rate that the material is degraded.

In our experiments, the material appeared encapsulated in the three analysis periods. Some researchers have found that PLA applications stimulate a foreign body reaction in subdermal implants in rats with fibrous capsule formation [57,58]. In the histological study, it was also observed that in the last observation period (90 days), the persistent material was fragmented with infiltration of inflammatory cells and a network of collagen fibers surrounding the material, similar to what was reported by García et al. [61].

The high persistence of nanocomposites in the subdermal tissue was due to the high crystallinity of the PLA, which was difficult for cell adhesion and penetration for the phagocytic process, causing slow degradation and reabsorption. It has been reported that some implants made of this material stimulate chronic inflammation and foreign body reactions with the presence of inflammatory infiltrate, fibrous capsules, and capillaries [62,63], as evidenced here. However, with the n-BGs incorporation, the reabsorption capacity increased. The increased heterogeneity and lower crystallinity facilitated the process (Figure 8 and Figure 9), as evidenced in the histological analysis of the nanocomposites (Figure 15).

In contrast to the three experimental formulations, the behavior of the positive control material (porcine collagen) was different since there was no fibrous capsule formation. Still, an inflammatory infiltrate is responsible for reabsorbing the biomaterial that will progressively decrease at 60 and 90 days until it is practically negligible. This resolution process in which there is no foreign body response with capsule formation is explained because collagen is a material of natural origin, a fundamental component of the extracellular matrix of all tissues, and highly biocompatible [64].

Not many scaffolds can simultaneously stimulate bone cancer control and cell regeneration with new cells [65] Thus, there is an urgent need to develop compatible and persistent materials, including those with therapeutic effects. The histological analyses of our nanocomposites showed that they are materials with a slow reabsorption process but are biocompatible and thus beneficial for the regeneration of tissues that require much slower healing processes (bone tissue). They can also be helpful as drug release agents to control diseases such as bone cancer while osteogenesis is stimulated. Therefore, producing this type of persistent material that could fulfill this dual purpose is very promising for biomedical applications in future research. 

## 3. Materials and Methods

Tetraethoxysilane, calcium nitrate, nitric acid, ammonium, chloroform, and ammonium phosphate salts were obtained from Sigma Aldrich (Palo Alto, CA, USA). Polypropylene glycol (PEG, Mn 16,000–24,000 Da) and dioxane were supplied from Fluka (Seelze, Lower Saxony, Germany). PLA with an Mn 200,000 Da containing 1.5–2% of the D isomer (PLA-2002D grade) was obtained from Nature Works LLC, USA (NatureWorks, Minnetonka, MN, USA).

### 3.1. Synthesis of n-BGs

An n-BG based on a ternary system with an approximate molar composition of 54%-SiO_2_:40%-CaO:6%-P_2_O_5_ was synthesized as previously reported with some modifications [5]: 

Two solutions were used to prepare the n-BGs: the first solution consisted of adding 7.7 g of Ca(NO_3_)_2_∙3H_2_O in 117 mL of distilled water. The second solution consisted of 9.7 mL of tetraethoxysilane (TEOS) in 63.5 mL of ethanol. Finally, the solutions were mixed in a 500 mL beaker, maintaining the pH between 1-2 with citric acid.

The resulting solution was dropped under constant stirring onto a solution of 1.2 g of NH_4_H_2_PO_4_ in 1500 mL of distilled water, adjusting the pH to 10 by adding aqueous ammonia. The resulting mixture was stirred for four days. The residue obtained was centrifuged with an Eppendorf Centrifuge 5810 (Hamburg, Germany) and then washed through three centrifugation cycles with distilled water. The solid was dispersed in 200mL of a 2 *w*/*v*% aqueous polypropylene glycol solution and stirred for 24 h. This suspension was frozen in an Eppendorf ultra-freezer, Innova^®^ U101 (Hamburg, Germany) at −80 °C for two hours, then lyophilized in a Christ, Alpha 2-4 LD plus freeze-dryer (Osterode am Harz, Germany) for 48 h and, finally, subjected to calcination at 350 °C for three hours.

#### 3.1.1. Characterization of the n-BGs

##### Fourier Transform Infrared Spectroscopy (FT-IR)

The n-BGs were chemically characterized by Fourier transform infrared spectroscopy (FT-IR-8400) in an IR affinity-1 infrared spectrophotometer (Shimadzu, Kyoto, Japan) between 500–4000 cm^−1^ using the transmittance mode with the diamond point method.

##### Thermal Analysis

The thermal degradation of the n-BGs was carried out on a NETZSCH TG 209 F1 Libra instrument (Mettler Toledo, Schwerzenbach, Switzerland). The sample was placed in alumina crucibles and heated to 900 °C at a heating rate of 10 °C/min. 

##### X-ray Diffraction (XRD)

X-ray diffractometry (XRD) experiments were performed in a PANalytical X′Pert PRO diffractometer (Malvern Panalytical), using Cu Kα1 radiation (1.540598 Å) and Kα2 (1.544426 Å) in a 2θ range between 5–60°.

##### Scanning Electron Microscopy (SEM)

The morphology of the n-BGs was analyzed at 20 kV in a scanning electron microscope (SEM) (Hitachi TM 3000, Musashino, Tokyo, Japan) with a secondary backscattered electron mode. For the correct conductivity of the samples, a gold coating was prepared.

##### Transmission Electron Microscopy (TEM)

A Philips Tecnai 12 transmission electron microscope (Mu-sashino, The Netherlands) was used to observe the size and shape of the n-BGs. The TEM sample was prepared by placing a drop of n-BGs/0.1% ethanol suspension on a TEM grid covered with a carbon film and evaporating the solvent entirely at room temperature.

### 3.2. Preparation of PLA/n-BGs scaffolds

Initially, a 5 *w*/*v*% homogenous solution of PLA was prepared in chloroform. Subsequently, concentrations of 5 *w*/*v*% and 10 *w*/*v*% of the n-BGs were added concerning the amount of PLA present in the solution, resulting in a translucent homogeneous solution of PLA/n-BGs. Finally, the mixture was left under stirring in an extraction hood until the total evaporation of the solvent.

#### 3.2.1. Characterization of PLA/n-BGs Nanocomposites

##### Fourier Transform Infrared Spectroscopy (FT-IR)

Employing an FT-IR, the functional groups found in the PLA/n-BGs nanocomposites were determined through an FT-IR instrument in the ATR (attenuated total reflectance) mode with a diamond tip (Shimadzu, Kyoto, Japan) between 500 and 4000 cm^−1^.

##### X-ray Diffraction (XRD)

The diffraction planes were elucidated through a PANalytical X’Pert PRO diffractometer (Malvern PANalytical, JarmanWay, Royston, UK) using Cu Kα1 radiation (1.540598 Å) and Kα2 (1.544426 Å), with a voltage accelerator of 45 kV, an electron generating current of 40 mA, an incident beam optical grating of 1°, and a diffracted beam grating of 9.1 mm in a spectral range of 2θ between 5–40°.

##### Scanning Electron Microscopy (SEM)

The morphology of the different nanocomposites was analyzed using a scanning electron microscope (SEM) (Hitachi TM 3000, Musashino, Tokyo, Japan). The microscope was used in the secondary mode of accelerated electrons for the analysis. The voltage used was 20 kV. A layer of gold-covered samples was used to improve the conductivity.

##### Transmission Electron Microscopy (TEM)

The size and shape of the different nanocomposites were analyzed by transmission electron microscopy at 80kV (TEM) (Philips Tecnai 12, Musashino, The Netherlands). The samples were prepared by adding 0.1% ethanol and were placed on a grid covered with a carbon film. Finally, the solvent was allowed to evaporate at room temperature. 

##### Thermal Analysis

The thermal properties of the PLA/n-BGs nanocomposites were measured by thermal gravimetric analysis (TGA) on a NETZSCH TG 209 F1 Libra (Mettler Toledo, Schwerzenbach, Switzerland). The sample was introduced into alumina crucibles and heated up to 900 °C at a heating rate of 10 °C/min under a nitrogen atmosphere (flow rate 50 mL/min). The glass transition temperature (T_g_), melting temperatures (T_m_), melting enthalpy (*Δ**H_m_*), cold crystallization temperature (T_cc_), and cold crystallization enthalpy (*Δ**H_CC_*) were determined by differential scanning calorimetry using a DSC1/500 (Mettler Toledo, Schwerzenbach, Switzerland). The samples were heated from 25–250 °C with a 10 °C/min ramp and a 60 mL/min nitrogen flow. TGA and DSC data were analyzed using TA Instruments Universal Analysis Software. The thermograms and values of thermal transitions were taken from the second scan to eliminate the thermal history of the polymer.

The percentage of crystallinity was calculated using the enthalpy of fusion of the PLA at 100% crystallinity ($\Delta H_{m}^{{}^{\circ}}$) through Equation (1) [65].
(1)$$X_{c}=\frac{\left(\Delta H_{m}-\Delta H_{CC}\right)}{\Delta H_{m}^{{}^{\circ}}\left(1-x\right)}$$
where $\Delta H_{m}^{{}^{\circ}}$ is 93 J/g, corresponding to the enthalpy of fusion of an ideal PLA at 100% crystallinity. $\Delta H_{m}$ and $\Delta H_{CC}$ are the fusion enthalpy and crystallization enthalpy of the nanocomposites in J/g, and *x* is the percentage of nanoparticles in the matrix polymer.

### 3.3. Biological Tests

#### 3.3.1. Analysis of Antimicrobial Activity

The inhibitory effect of the n-BGs was determined against *Escherichia coli* (ATCC 11775), *Vibrio parahaemolyticus* (ATCC 17802), *Staphylococcus aureus* subsp. aureus (ATCC 55804), and *Bacillus cereus* (ATCC 13061). The inocula were obtained from cultures in a liquid medium with 24 h of growth and adjusted to a concentration of 10^6^ CFU mL^−1^ using a spectrophotometer (Eppendorf BioSpectrometer^®^ kinetic, Hamburg, Germany) at 620 nm. Dilutions were made by washing cells, centrifuging the strains at 4000 rpm (Eppendorf Minispin, Hamburg, Germany), and resuspending the pellet obtained in peptone water by vortexing (IKA, Staufen, Germany).

For the test, the n-BGs were available in different concentrations (2.5, 5, 10, 15, and 20 wt.%). For this, 900 µL of nutrient broth, 100 µL of the suitably diluted strain, and the n-BGs required to adjust the desired concentration were placed in 1500 µL conical tubes. The conical tubes were stirred at 50 rpm in an incubation shaker (IKA, KS 3000 i control) at 37 °C for 24 h.

All analyses were performed in triplicate. To read the result, 40 µL of 2,3,5-triphenyltetrazolium chloride (TTC) at 0.2 wt.% was left in incubation for two hours. The red color produced by the dye indicated metabolic activity and microbial growth. To corroborate the negative results, 100 µL of the treatment was taken from each of the tubes and deposited in Petri dishes making serial dilutions with Müller Hinton agar and incubated at 37 °C for 24 h.

#### 3.3.2. MTT Cell Viability Assay

Preliminary cytotoxicity studies were performed using the salt (3-(4,5-dimethylthiazol-2-yl)-2,5-diphenyltetrazolium bromide) MTT, with the cell viability kit MTT cell proliferation assay kit (Abcam ab211091).

The protocol used for the test was the one recommended in the product’s technical datasheet. For this, 16 wells of 96 were used. In each well, 30,000 HeLa cells were placed with a complete culture medium (HBSS, 15% fetal bovine serum, antibiotics, and antimycotic) in an incubator with 5% CO_2_ for 24 h.

After 24 h, the culture medium was removed, the cells were washed with PBS, 50 µL of HBSS medium, and 50 µL of the MTT reagent were added. Next, the plate was incubated for three hours at 37 °C until intracellular formazan crystals were visible under a Leica DM750 light microscope (Leica Microsystems, Mannheim, Germany). Subsequently, the MTT kit solvent was added and shaken in an orbital shaker for 15 min, and the absorbance measurement at 590 nanometers was performed.

For each sample, three replicates were performed, including positive and negative controls, and two successive readings were taken. Cisplatin was used as the positive control and the cell culture medium as the negative control. To determine the viability or biocompatibility percentage, it is assumed that the absorbance is proportional to the number of viable cells, according to Equation (2) of the cell MTT viability percentage [66]: % Cell viability MTT = Absorbance sample/Absorbance untreated × 100 (2)

#### 3.3.3. Histological Analysis of PLA/n-BGs Nanocomposites

The objective of this part of the study was to describe the tissue changes that occurred due to subcutaneous implantation in nine Wistar rats. The biomodels were male, five months old, with an average weight of 380 g, implanted with three formulations of the nanocomposite (PLA, PLA/n-BGs 5 wt.%, and PLA/n-BGs 10 wt.%). As a positive control for resorption, samples of a porcine collagen membrane were implanted.

The biomodels were distributed in three groups, making 30, 60, and 90 day observations. The animals were housed in the LABBIO laboratory of the Universidad del Valle in Cali, Colombia, with water and food.

After implantation, the biomodels were euthanized with sodium pentobarbital 390 mg/mL, sodium diphenylhydantoin 50 mg/mL (Euta-nex^®^, Invet, Medellin, Colombia) at an intraperitoneal dose of 0.3 mL per kg of weight.

Once the biomodels were euthanized, a trichotomy of the implantation area was performed. Subsequently, the implanted samples were recovered, which were fixed in buffered formalin for 48 h (100 mL of 37% formalin, 900 mL of distilled water, 0.4 mL of monobasic sodium phosphate, and 0.65 mL of dibasic sodium phosphate) in a ratio volume (10:1 sample) and were washed with a buffer solution under gentle agitation for 15 min.

For processing the samples (dehydration with alcohols, diaphanization with xylol, and infiltration with paraffin), a Leica^®^ TP1020^®^ automatic tissue processor (Leica Microsystems, Mannheim, Germany) was used. Subsequently, the samples were embedded in paraffin (Paraplast Xtra^®^ McCormick^®^) and cut with a thickness of 6 µm with the Leica microtome (Leica Microsystems, Mannheim, Germany).

Hematoxylin-Eosin staining and Masson’s Trichrome were used to perform the histological analysis. Once the samples were stained, photographs were taken with a Leica microscope (Leica Microsystems, Mannheim, Germany), and the images were analyzed.

### 3.4 Statistic Analysis

Statistical differences for mechanical properties of the nanocomposites and MTT test were evaluated by analysis of variance (ANOVA) using Minitab software (2019). The Fisher and Dunnett test were used as post-ANOVA methods to compare and establish the differences with a significance of 5% (*p* < 0.05).

## 4. Conclusions

In this work, we synthesized and characterized bioglass nanoparticles with a size of ca. 24.87 ± 6.26 nm and with the ability to distribute themselves in the PLA polymeric matrix. These nanoparticles presented the characteristic FTIR bands of these ceramic compounds and their crystalline planes by XRD. Additionally, they showed a sub-lethal antibacterial effect against *Vibrio parahaemolyticus* (ATCC 17802), *Staphylococcus aureus* subsp. aureus (ATCC 55804), and *Bacillus cereus* (ATCC 13061), and an inhibitory effect of *Escherichia coli* (ATCC 11775) at 20 *w*/*v*%.

With the nanoparticles, PLA/n-BGs nanocomposites were obtained at 5 wt.% and 10 wt.%, as evidenced by FT-IR spectroscopy. The nanocomposites presented a sponge-like morphology characterized by heterogeneity, thanks to the incorporation of the n-BGs. Additionally, the incorporation of the n-BGs at 5 wt.% and at 10 wt.% reinforced the mechanical properties of the PLA by increasing Young’s modulus by ca. 91%. However, by incorporating n-BGs, the percentage of the PLA crystallinity decreased. That helped explain why these nanocomposites were resorbed in a higher ratio than the PLA scaffolds.

On the other hand, the incorporation of the n-BGs into the PLA did not affect the cell viability of HeLa cells since the percentage of MTT viability remained above 80% during the tests. In the case of subdermal implantation analyses in biomodels, reabsorption was very slow, showing a significant presence in the material even three months after implantation, with an inflammatory response and an inflammatory capsule. The addition of the n-BGs to the nanocomposites decreased the crystallinity while increasing the material’s heterogeneity. Those changes helped the nanocomposites have higher reabsorption in the subdermal tissues than the neat PLA without affecting their biocompatibility. These results demonstrate the potential use of our nanocomposites in tissue regeneration with slow healing processes (bone tissue) or as a drug release agent. 

## Figures and Tables

**Figure 1 molecules-27-03640-f001:**
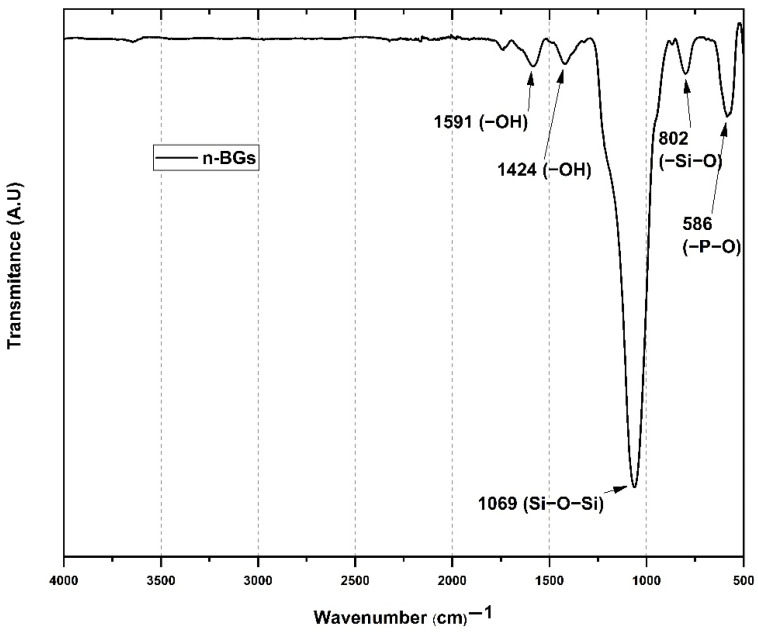
FT−IR spectrum for the n−BGs.

**Figure 2 molecules-27-03640-f002:**
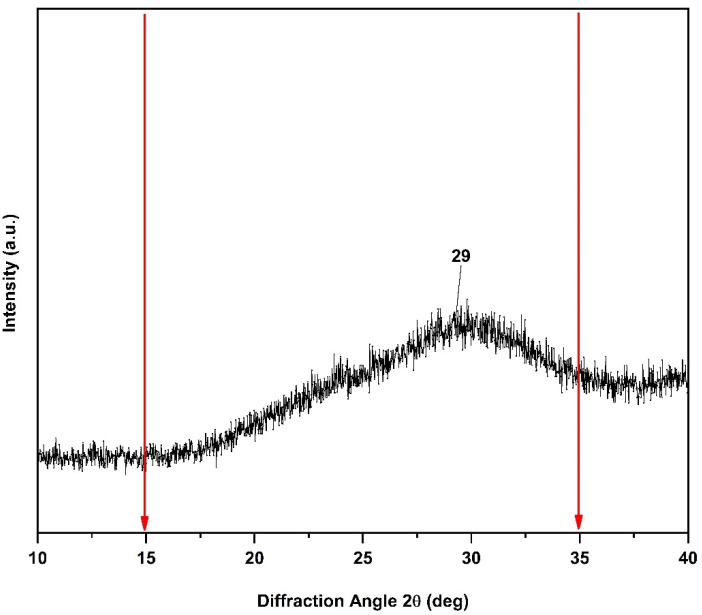
XRD patterns of the n-BGs.

**Figure 3 molecules-27-03640-f003:**
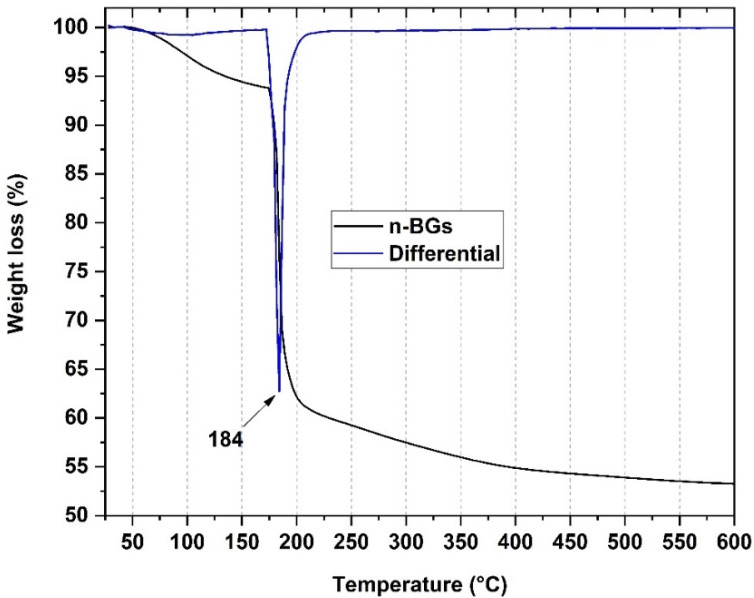
Thermogravimetric analysis of the n-BGs.

**Figure 4 molecules-27-03640-f004:**
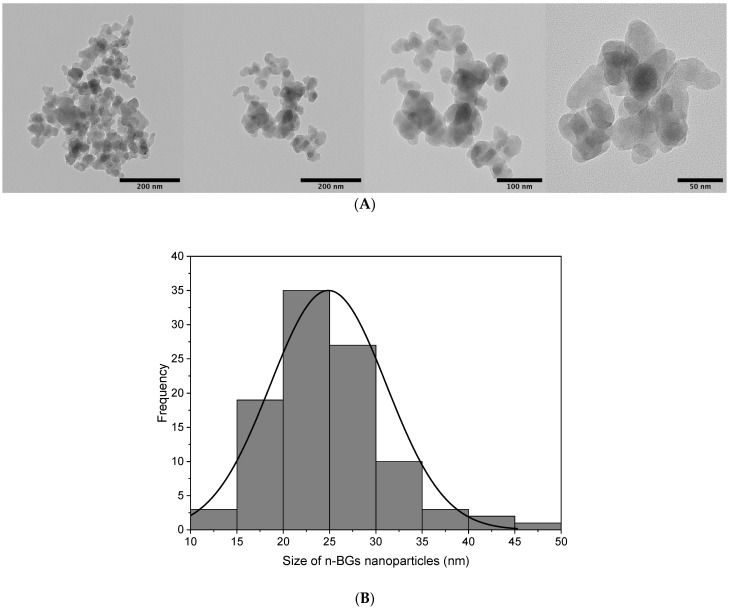
(**A**) TEM images of the n-BGs at 50, 100, and 200 nm and (**B**) histogram of the n-BGs.

**Figure 5 molecules-27-03640-f005:**
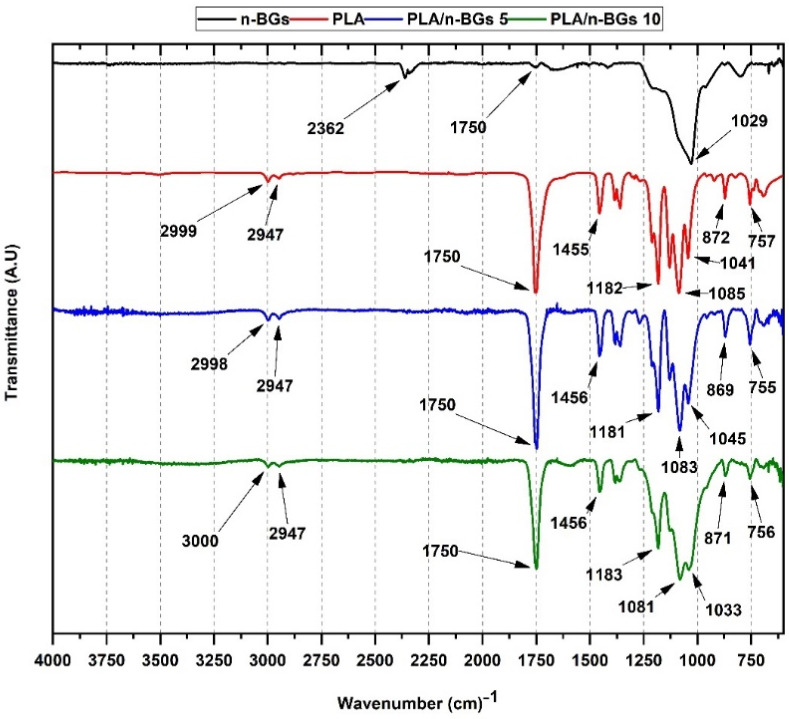
FT-IR of neat PLA and PLA/n-BGs nanocomposites of 5 wt.% and 10 wt.%.

**Figure 6 molecules-27-03640-f006:**
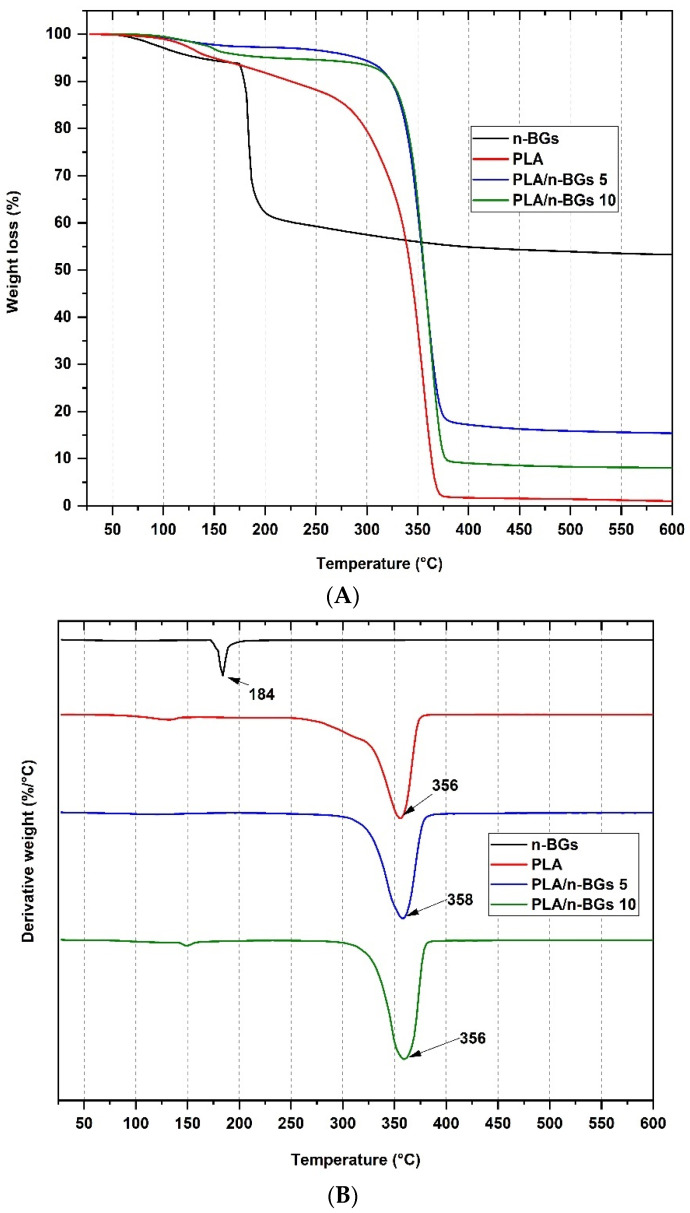
TGA and DTGA analysis of the nanocomposites of (**A**) neat PLA, PLA/n-BGs 5 wt.%, and PLA/n-BGs 10 wt.%. (**B**) Derivative TGA of neat PLA, PLA/n-BGs 5 wt.%, and PLA/n-BGs 10 wt.%.

**Figure 7 molecules-27-03640-f007:**
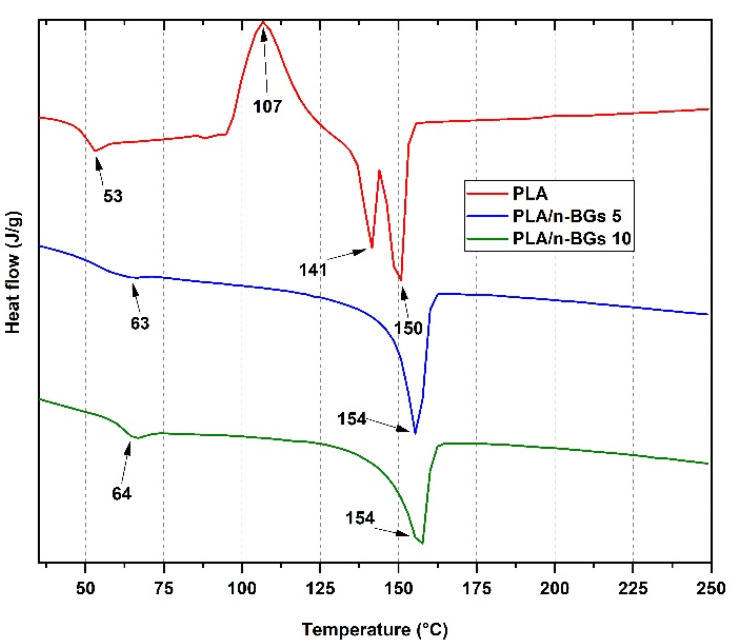
DSC thermograms for the neat PLA nanocomposites, PLA/n-BGs 5 wt.%, and PLA/n-BGs 10 wt.%.

**Figure 8 molecules-27-03640-f008:**
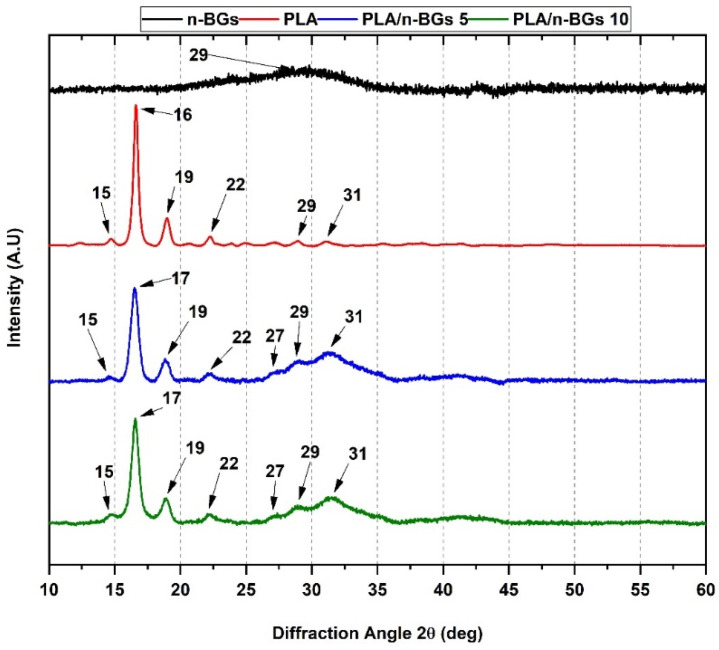
XRD analysis of neat PLA, PLA/n-BGs 5 wt.% and PLA/n-BGs 10 wt.%.

**Figure 9 molecules-27-03640-f009:**
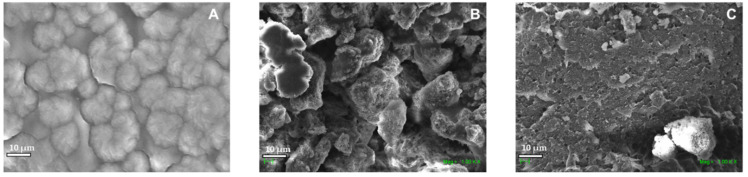
Electron micrographs of (**A**) neat PLA, (**B**) PLA/n-BGs 5 wt.%, and (**C**) PLA/n-BGs 10 wt.% at 1000×.

**Figure 10 molecules-27-03640-f010:**
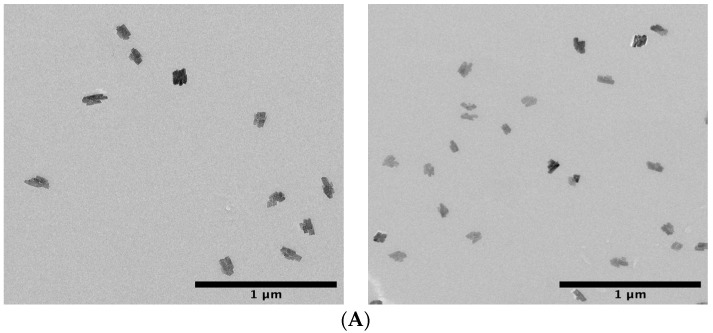
TEM images of PLA/n-BGs nanocomposites: (**A**) PLA/n-BGs 5 wt.% and (**B**) PLA/n-BGs 10 wt.%.

**Figure 11 molecules-27-03640-f011:**
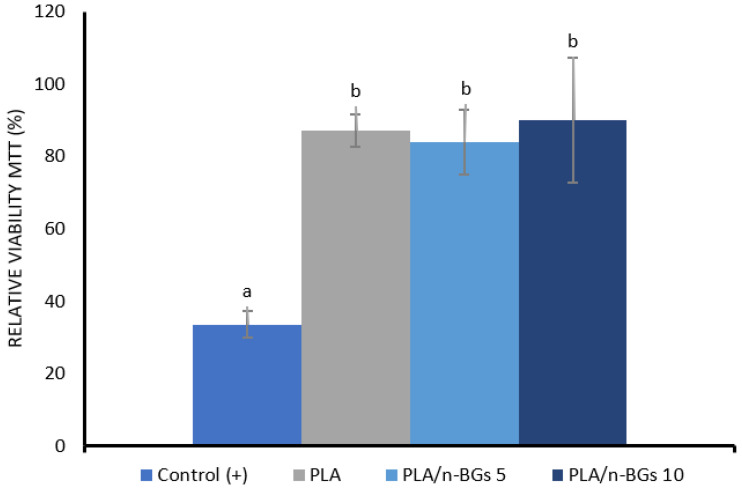
MTT data for the different formulations of neat PLA and PLA/n-BGS nanocomposites.

**Figure 12 molecules-27-03640-f012:**
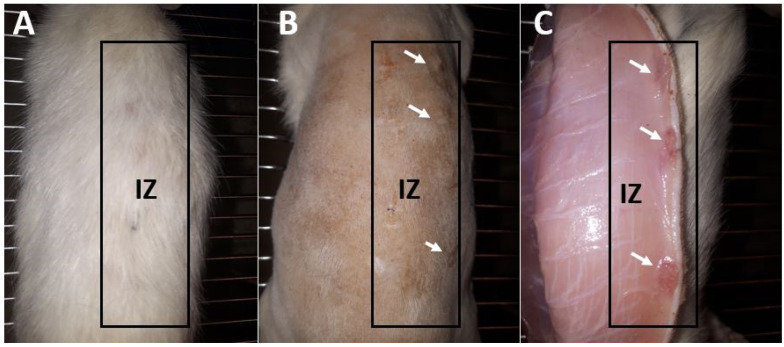
Macroscopic image of a biomodel implanted at 30 days. (**A**). Dorsal image. (**B**) Dorsal image with a trichotomy. (**C**) Inner surface of the skin. IZ: Implantation zone. White arrows: Implantation sites.

**Figure 13 molecules-27-03640-f013:**
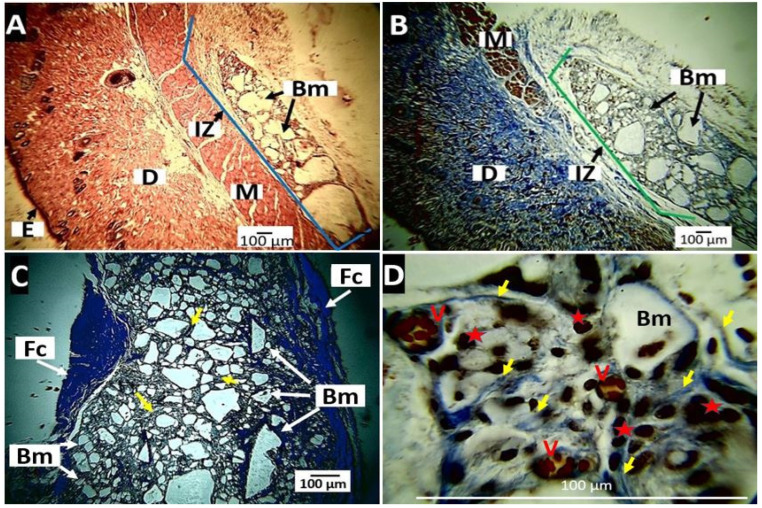
Neat PLA implanted at 30, 60, and 90 days. (**A**): 30 days, Hematoxylin-Eosin technique, 4×. (**B**): 60 days, Masson’s Trichrome technique, 4×. (**C**): 90 days, Masson’s Trichrome technique, 10×. (**D**): 30 days, Masson’s Trichrome technique, 100×. E: epidermis. D: dermis. M: muscle. IZ: implantation zone. Bm: biomaterial. Fc: fibrous capsule. Yellow arrows: collagen I fiber. V: blood vessel. Red star: inflammatory cells.

**Figure 14 molecules-27-03640-f014:**
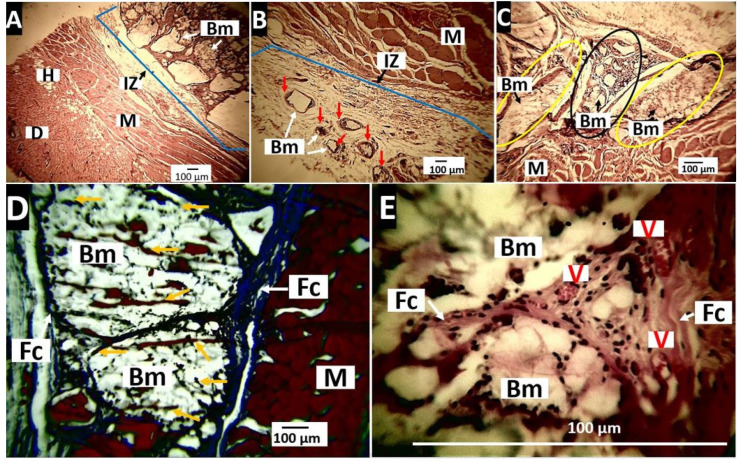
Nanocomposites PLA/n-BGs 5 wt.% implanted at 30, 60, and 90 days. (**A**): 30 days, Hematoxylin-Eosin technique, 4×. (**B**): 60 days, Masson’s Trichrome technique, 4×. (**C**): 90 days, Masson’s Trichrome technique 10×. Black oval: particles of biomaterial in the process of fragmentation. Yellow oval: biomaterial particles with surface changes in the process of degradation. (**D**): 60 days, Masson’s Trichrome technique, 10×. (**E**): 60 days, Hematoxylin Eosin technique 100×. D: dermis. H: hypodermis. M: muscle. IZ: implantation zone. Bm: biomaterial. Fc: fibrous capsule. Red arrows: particles of the material in the process of degradation/reabsorption. Yellow arrows: collagen I fiber. V: blood vessel.

**Figure 15 molecules-27-03640-f015:**
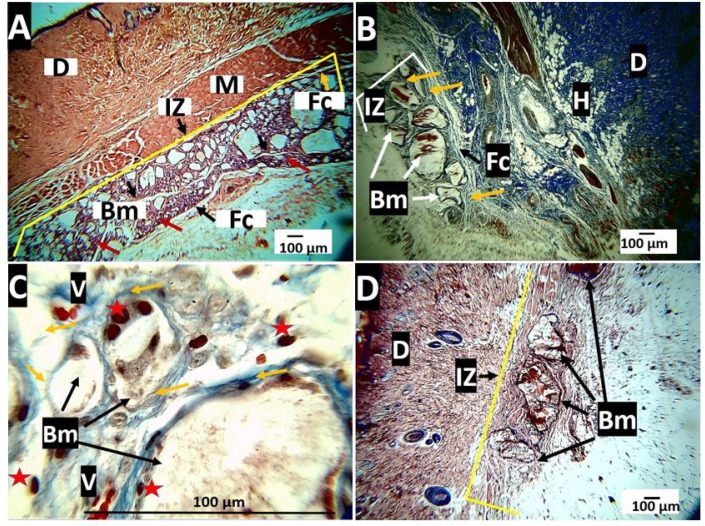
Nanocomposites of PLA/n-BGs 10 wt.% implanted at 30, 60, and 90 days. (**A**): 30 days, Hematoxylin-Eosin technique, 4×. (**B**): 60 days, Masson’s Trichrome technique, 4×. (**C**): 60 days, Masson’s Trichrome technique, 100×. (**D**): 90 days, hematoxylin-eosin technique, 4×. D: dermis. H: hypodermis. M: muscle. IZ: implantation zone. Fc: fibrous capsule. Bm: biomaterial. Yellow arrows: collagen I fiber. V: blood vessel. Red arrows: inflammatory cells. Red star: inflammatory cells.

**Figure 16 molecules-27-03640-f016:**
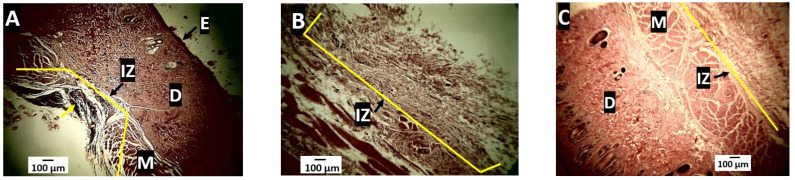
Porcine collagen samples implanted at 30, 60, and 90 days. (**A**): 30 days, Hematoxylin-Eosin technique, 4×. (**B**): 60 days, Hematoxylin-Eosin technique, 4×. (**C**): 90 days, hematoxylin-eosin technique, 4×. E: epidermis. D: dermis. M: muscle. IZ: implantation zone.

**Table 1 molecules-27-03640-t001:** Thermal properties of neat PLA, PLA/n-BGs 5 wt.%, and PLA/n-BGs 10 wt.%.

Sample	Tg (°C)	T_CC_ (°C)	T_m1_ (°C)	T_m2_ (°C)	Χc (%)	T10 (°C)	Tmax (°C)
Neat PLA	53	107	141	150	7.8	224	356
PLA/n-BGs 5	63	NP	NP	154	5.0	323	358
PLA/n-BGs 10	64	NP	NP	154	5.8	323	356

T_g_: glass transition, T_cc_: cold crystallization temperature, T_m1_,T_m2_: melting temperatures, Χ_c_: crystallinity percentage, T_10_: decomposition temperature at 10% weight loss, and T_max_: temperature for maximum weight loss rate. NP: not presented.

**Table 2 molecules-27-03640-t002:** Mechanical properties for n-BGs.

Sample	Young’s Modulus (N/mm^2^) (MPa) *	Compression Max (MPa) *	Deformation Max (%) *
Neat PLA	1.49 ^b^ ± 0.44	1.13 ^a^ ± 0.02	123.79 ^b^ ± 50.30
PLA/n-BGs 5	2.85 ^a^ ± 0.76	0.99 ^a^ ± 0.15	38.46 ^a^ ± 0.56
PLA/n-BGs 10	2.19 ^ab^ ± 0.33	0.87 ^a^ ± 0.17	62.37 ^b^ ± 9.61

* Different letters in the same column indicate significant differences (*p* < 0.05).

**Table 3 molecules-27-03640-t003:** Inhibition of n-BGs against bacterial strains.

Strains	Control	2.5 *w*/*v*%	5 *w*/*v*%	10 *w*/*v*%		15 *w*/*v*%		20 *w*/*v*%	
	TTC	TTC	TTC	TTC	Log CFU mL ^−1^	TTC	Log CFU mL ^−1^	TTC	Log CFU mL ^−1^
*Bacillus cereus*	+++	+	+	−	1.3 ± 0.3	−	1.1 ± 0.1	−	0
*Staphylococcus aureus*	+++	+	+	+		−	1.7 ± 0.4	−	0.5 ± 0.1
*Escherichia coli*	+++	+	+	+		−	1.4 ± 0.3	−	0.3 ± 0.1
*Vibrio parahaemolyticus*	+++	++	+	+		+		−	0.3 ± 0.2

For results with no pathogen activity, the concentration of the pathogen is shown (Log CFU mL^−1^). (+++) Strong activity of the pathogen; (++) moderated activity of the pathogen; (+) weak activity of the pathogen; (−) pathogen without activity. TTC: 2,3,5-triphenyltetrazolium chloride.

## Data Availability

Data are available under request to the corresponding author.

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
