# Peer review of "Biocompatibility Assessment of Polylactic Acid (PLA) and Nanobioglass (n-BG) Nanocomposites for Biomedical Applications"

_molecules, 2022, doi:10.3390/molecules27113640_

Round 1

Reviewer 1 Report

The aim of this work was to produce a persistent material with a dual purpose - to be biocompatible and to be able of drug release. These aims were fulfilled, which are very promising. The results are significant and novel.

The manuscript is clear and well-structured. References are relevant, nevertheless few of them could be omitted, especially the older ones. 

The numbering of chapters is not right and it is not complete:

l. 168 and l. 412 - the same number 2.2 with different content...and more, it is necessary to renumber.

l. 29, 761, 841-842 should be - Staphylococcus aureus subsp. aureus

l. 394-395 - it is not necessary to be in italics - gram-positive, gram-negative

Figure 11 is not very useful.

Author Response

Reviewer 1

The aim of this work was to produce a persistent material with a dual purpose - to be biocompatible and to be able of drug release. These aims were fulfilled, which are very promising. The results are significant and novel.

R// We greatly appreciate the positive comments from the reviewer about the goal of our work.

The manuscript is clear and well-structured. References are relevant, nevertheless few of them could be omitted, especially the older ones.

R// We appreciate the suggestion from the reviewer. Some references from the introduction section were updated (blue color) and some were deleted for a better understanding of the background.

The numbering of chapters is not right and it is not complete: l. 168 and l. 412 - the same number 2.2 with different content...and more, it is necessary to renumber.

R// We appreciate the suggestion from the reviewer. The proper correction was done and labeled with blue.

  1. 29, 761, 841-842 should be - Staphylococcus aureus subsp. Aureus

R// We appreciate the suggestion from the reviewer. The proper correction was done and labeled with blue.

  1. 394-395 - it is not necessary to be in italics - gram-positive, gram-negative

R// We appreciate the suggestion from the reviewer. The proper correction was done and labeled with blue.

Figure 11 is not very useful.

R// We appreciate the suggestion from the reviewer. However, since we used the data from the table for the discussion of the cell viability MTT, we decided not to remove the figure. MDPI journals do not have a limit of figures and pages in the manuscript. For instance, we conclude that removing the figure to add it to supporting information might be unnecessary in this case. Otherwise, if the reviewer indicates to us that the section is not correct or needs to be completely removed

Reviewer 2 Report

In the present work, characterization, and assessment of the biocompatibility of polylactic acid (PLA) and nano- 2 bioglass (n-BG) nanocomposites were realized for biomedical applications.

The study is very well organized, and all the required experiments were performed to characterize the novel composite biomaterial. In addition, the required tests were realized to test the biocompatibility of nanocomposite material. 

This study is novel and promising for the community and I strongly recommend its publication.

I congratulate all the authors for their work.

I have only a few minor suggestions below:

Please use "histological analyses" instead of histological analyzes throughout the MS. Line 32-644

Line 773: Use "analyses" instead of analyzes.

I also recommend the authors use 2 significant figures after the decimal point. Line 27: 24.87 ± 6.26

Author Response

In the present work, characterization, and assessment of the biocompatibility of polylactic acid (PLA) and nano- 2 bioglass (n-BG) nanocomposites were realized for biomedical applications. The study is very well organized, and all the required experiments were performed to characterize the novel composite biomaterial. In addition, the required tests were realized to test the biocompatibility of nanocomposite material. This study is novel and promising for the community and I strongly recommend its publication. I congratulate all the authors for their work.

R// We greatly appreciate the positive comments from the reviewer about the novelty, structure, and experiments in our work.

I have only a few minor suggestions below: Please use "histological analyses" instead of histological analyzes throughout the MS. Line 32-644

R// We appreciate the suggestion from the reviewer. The proper correction was done and labeled with blue.

Line 773: Use "analyses" instead of analyzes.

R// We appreciate the suggestion from the reviewer. The proper correction was done and labeled with blue.

I also recommend the authors use 2 significant figures after the decimal point. Line 27: 24.87 ± 6.26

R// We appreciate the suggestion from the reviewer. The proper correction was done and labeled with blue.
